# A Nanosensor for Naked-Eye Identification and Adsorption of Cadmium Ion Based on Core–Shell Magnetic Nanospheres

**DOI:** 10.3390/ma13173678

**Published:** 2020-08-20

**Authors:** Yaohui Xu, Chi Deng, Zhigang Xiao, Chang Chen, Xufeng Luo, Yang Zhou, Qiang Jiang

**Affiliations:** 1Laboratory for Functional Materials, School of Electronics and Materials Engineering, Leshan Normal University, Leshan 614000, China; xyh1986@lsnu.edu.cn (Y.X.); dengchi@lsnu.edu.cn (C.D.); xiaozhigang@lsnu.edu.cn (Z.X.); chenchang@lsnu.edu.cn (C.C.); luoxufeng@lsnu.edu.cn (X.L.); 2School of Textile Science and Engineering, National Engineering Laboratory for Advanced Yarn and Clean Production, Wuhan Textile University, Wuhan 430200, China

**Keywords:** Fe_3_O_4_@SiO_2_, magnetism, fluorescence, adsorption, Cd^2+^

## Abstract

Fe_3_O_4_@SiO_2_ nanospheres with a core–shell structure were synthesized and functionalized with bis(2-pyridylmethyl)amine (BPMA). The photoresponses of the as-obtained Fe_3_O_4_@SiO_2_-BPMA for Cr^3+^, Cd^2+^, Hg^2+^ and Pb^2+^ ions were evaluated through irradiation with a 352 nm ultraviolet lamp, and Fe_3_O_4_@SiO_2_-BPMA exhibited remarkable fluorescence enhancement toward the Cd^2+^ ion. The adsorption experiments revealed that Fe_3_O_4_@SiO_2_-BPMA had rapid and effective adsorption toward the Cd^2+^ ion. The adsorption reaction was mostly complete within 30 min, the adsorption efficiency reached 99.3%, and the saturated adsorption amount was 342.5 mg/g based on Langmuir linear fitting. Moreover, Fe_3_O_4_@SiO_2_-BPMA displayed superparamagnetic properties with the saturated magnetization of 20.1 emu/g, and its strong magnetic sensitivity made separation simple and feasible. Our efforts in this work provide a potential magnetic functionalized nanosensor for naked-eye identification and adsorption toward the Cd^2+^ ion.

## 1. Introduction

Cadmium (Cd) is an important rare element, which is widely used in the production of pigments [1], phosphors [2] and photocells [3]. Because of its strong toxicity, the Cd^2+^ ion can cause serious environmental and health issues, including itai-itai disease, lung cancer, renal cancer and prostatic cancer [4,5,6]. Therefore, some fluorescent chemosensors have been developed for the selective recognition and sensing of the Cd^2+^ ion. These chemosensors can be induced by the Cd^2+^ ion and exhibit naked-eye changes in fluorescence. High sensitivity, high selectivity, instantaneous response and simplicity are the greatest advantages of these probe molecules [7,8,9,10]. However, these chemosensors for Cd^2+^ are meant to be disposable and for single use only, and they are difficult if not impossible to recycle. On this basis, the incorporation of fluorescent chemosensors and nanomaterials into an inorganic/organic hybrid nanocomposite is presented.

Magnetic Fe_3_O_4_@SiO_2_ nanoparticles with a core–shell structure have attracted an increasing interest in biological and environmental fields [11,12,13,14]. The particles with nano size possess abundant surface active sites for surface modification, and the greatest strength of Fe_3_O_4_@SiO_2_ nanoparticles is their targeting ability. These magnetic particles can be attracted to a target zone under the action of an external magnetic field, which makes recovery feasible [15,16,17,18,19,20,21]. Moreover, the inert SiO_2_-shell can prevent the magnetic Fe_3_O_4_-core from oxidation, and the abundant surface hydroxyl of SiO_2_ is highly conducive to surface grafting with organic ligands [22,23,24,25].

In this work, we developed an inorganic/organic hybrid nanosensor for Cd^2+^ detection, in which the core–shell Fe_3_O_4_@SiO_2_ nanospheres were employed as a matrix, and bis(2-pyridylmethyl)amine (BPMA) was grafted onto the end of the Fe_3_O_4_@SiO_2_ surface through the “grafting from” method. BPMA is a tridentate ligand with two terminal nitrogen-donor coordination sites (pyridyl) and the central nitrogen donor site (amine), which can coordinate with some heavy metal ions to form complexes. The coordination behavior with metal ions caused an interesting photoresponse, especially ones with d^10^ metal centers. The as-obtained Fe_3_O_4_@SiO_2_-BPMA showed a significant change after binding with a Cd^2+^ ion in the excitation of ultraviolet light, which was visible to the naked eye. Moreover, Fe_3_O_4_@SiO_2_-BPMA also showed rapid and efficient adsorption toward the Cd^2+^ ion. It is worth noting that Fe_3_O_4_@SiO_2_-BPMA possesses superparamagnetism, and the strong magnetic sensitivity makes recycling easier.

## 2. Experimental

### 2.1. Materials

FeCl_2_**∙**4H_2_O, FeCl_3_**∙**6H_2_O, NH_3_**∙**H_2_O (25 wt.%), trisodium citrate and ethyl silicate were purchased from Shanghai Macklin Biochemical Co. Ltd. (Shanghai, China) Trichloromethylsilane, trichloro(3-chloropropyl)silane, branched polyethylenimine (Mw ~25,000), 4-bromonaphthalic anhydride and bis(2-pyridylmethyl)amine (BPMA) were purchased from Sigma-Aldrich Co. Ltd. (Shanghai, China)

### 2.2. Synthesis of Fe_3_O_4_@SiO_2_ Nanospheres

Firstly, Fe_3_O_4_ particles were synthesized based on a chemical co-precipitation strategy. Briefly, FeCl_2_**∙**4H_2_O (10 mmol) and FeCl_3_**∙**6H_2_O (20 mmol) were dissolved with distilled water (120 mL), and 60 mL of NH_3_**∙**H_2_O (25 wt.%) was added to the above solution under a N_2_ atmosphere with mechanical stirring. The mixture was kept for 30 min at 70 °C. After that, the black precipitate was collected and washed with distilled water. Subsequently, the as-synthesized Fe_3_O_4_ particles were added to a trisodium citrate solution (150 mL, 20 mmol/L), the mixture was kept with mechanical stirring for 12 h at room temperature, and N_2_ was bubbled throughout the reaction. After that, the citrate-capped Fe_3_O_4_ was washed with distilled water and ethanol. 

Then, Fe_3_O_4_@SiO_2_ particles were synthesized using a sol–gel process by the hydrolysis and condensation of ethyl silicate on the Fe_3_O_4_ seed. Briefly, the citrate-capped Fe_3_O_4_ (0.56 g), NH_3_∙H_2_O (5.0 mL, 25 wt.%) and ethyl silicate (4.0 mL) were added to ethanol (120 mL) under mechanical stirring, and the mixture was kept for 8 h at room temperature under a N_2_ atmosphere. After that, the as-synthesized Fe_3_O_4_@SiO_2_ precipitate was collected and washed with ethanol.

### 2.3. Fe_3_O_4_@SiO_2_ Functionalized with BPMA

The surface modifications on Fe_3_O_4_@SiO_2_ are illustrated by a flow chart in Figure 1. Firstly, the Fe_3_O_4_@SiO_2_ surface was functionalized with chlorine groups (labeled as Fe_3_O_4_@SiO_2_-Cl). Fe_3_O_4_@SiO_2_ (0.60 g), trichloromethylsilane (17.6 mmol) and trichloro(3-chloropropyl)silane (1.4 mmol) were added to hexane (150 mL), and the mixture was kept for 24 h with mechanical stirring at room temperature in an atmosphere of hydrogen chloride. After that, the as-synthesized Fe_3_O_4_@SiO_2_-Cl particles were washed with ethanol.

Secondly, the Fe_3_O_4_@SiO_2_ surface was functionalized with amino groups (labeled as Fe_3_O_4_@SiO_2_-NH_2_). The above Fe_3_O_4_@SiO_2_-Cl, methanol (5.0 mL) and branched polyethylenimine (1.0 mL) were added to distilled water (120 mL) under a N_2_ atmosphere with mechanical stirring, and the mixture was kept for 48 h at 65 °C. After that, the as-synthesized Fe_3_O_4_@SiO_2_-NH_2_ particles were washed with distilled water and ethanol.

Thirdly, the Fe_3_O_4_@SiO_2_ surface was functionalized with bromine groups (labeled as Fe_3_O_4_@SiO_2_-Br). The above Fe_3_O_4_@SiO_2_-NH_2_ and 4-bromonaphthalic anhydride (5.4 mmol) were added to ethanol (80 mL) under a N_2_ atmosphere with mechanical stirring, and the mixture was refluxed for 36 h in darkness. After that, the as-synthesized Fe_3_O_4_@SiO_2_-Br particles were washed with ethanol.

Finally, the Fe_3_O_4_@SiO_2_ surface was functionalized with bis(2-pyridylmethyl)amine (labeled as Fe_3_O_4_@SiO_2_-BPMA). The above Fe_3_O_4_@SiO_2_-Br and BPMA (7.02 mmol) were added to toluene (120 mL) under a N_2_ atmosphere with mechanical stirring, and the mixture was refluxed for 36 h in darkness. After that, the as-synthesized Fe_3_O_4_@SiO_2_-BPMA particles were washed with ethanol and then dried in vacuum at 60 °C for 24 h.

### 2.4. Characterization

X-ray diffraction (XRD) analyses were carried out using a DX-2700 instrument (Dandong, China) with Cu Kα radiation (30 kV, 25 mA). Transmission electron microscopy (TEM) and scanning electron microscopy (SEM) images were obtained by means of JEM-2100 (Tokyo, Japan) and SU8020 (Tokyo, Japan) instruments, respectively. Fourier transform infrared (FTIR) spectra were obtained using a Spectrum One (Nicolet iS 10, Waltham, MA, USA). X-ray photoelectron spectroscopy (XPS) analyses were performed by an ESCALAB 250Xi electron spectrometer (Waltham, MA, USA). Magnetization curves were obtained using a vibrating sample magnetometer (VSM, Lakeshore 7307, Columbus, OH, USA).

### 2.5. Adsorption Study

The adsorption capacities of Fe_3_O_4_@SiO_2_-BPMA were evaluated by removing heavy metal ions with strong toxicity (Cr^3+^, Cd^2+^, Hg^2+^ and Pb^2+^ ions) from simulated wastewaters. Briefly, 50 mg Fe_3_O_4_@SiO_2_ or Fe_3_O_4_@SiO_2_-BPMA powders were dispersed in 50 mL of the simulated wastewaters without pH pre-adjustments. The mixture was stirred with a constant speed at room temperature. Then, a certain amount of suspension was withdrawn at regular intervals, and the qualitative analysis of metal ions was performed. The adsorption efficiencies (*η*_t_, %) at a time t were calculated using Equation (1).
(1)$$\eta_{t}=\frac{C_{0}-Ct}{C_{0}}\times 100\;\%$$

## 3. Results and Discussion

Figure 2 shows the XRD patterns of Fe_3_O_4_ and Fe_3_O_4_@SiO_2_ powders. As observed, the XRD patterns of Fe_3_O_4_ showed several well-resolved peaks indexed to (220), (311), (400), (422), (511) and (440) planes, which are considered strongly consistent with the database of magnetite (JCPDS No. 65-3107). According to the Scherrer formula, the grain size of Fe_3_O_4_ was calculated, and the value was about 9.6 nm. For the XRD pattern of Fe_3_O_4_@SiO_2_, besides the diffraction peaks of the Fe_3_O_4_ phase, a broad and weak peak at 2*θ* = 11~27° (the orange dotted bordered rectangle) was observed, which was assigned to amorphous SiO_2_. This suggests that the Fe_3_O_4_ crystal was successfully composited with amorphous SiO_2_.

Figure 3a shows the TEM image of Fe_3_O_4_ particles. As observed, the Fe_3_O_4_ particles exhibited a mean size of about 10 nm, consistent with the calculation of XRD analysis. After compositing with SiO_2_, the size of particles increased to about 24.0 nm, as shown in the SEM image in Figure 3b, and the TEM image in Figure 3c reveals the obvious core–shell structure with a shell thickness of about 7.0 nm. This suggests that Fe_3_O_4_@SiO_2_ nanospheres with a core–shell structure were successfully synthesized using a sol–gel process by the hydrolysis and condensation of ethyl silicate on the Fe_3_O_4_ seed. However, the Fe_3_O_4_@SiO_2_ particles were aggregated slightly, which may be attributed to the enhancement of surface activity after coating SiO_2_ on the Fe_3_O_4_ seed because of more abundant surface hydroxyl from SiO_2_. Figure 3d,e show the SEM and TEM images of Fe_3_O_4_@SiO_2_-BPMA particles, respectively. There were no significant changes in either size or morphology compared with those of Fe_3_O_4_@SiO_2_ particles.

The grafted groups on the Fe_3_O_4_@SiO_2_ surface were characterized by FTIR. As observed in Figure 4a–e, the peaks for all samples at 3437, 1086 and 576 cm^−1^ were ascribed to the stretching vibration of O-H, Si-O and Fe-O bonds, further demonstrating the successful synthesis of Fe_3_O_4_@SiO_2_ with abundant surface hydroxyl. In the FTIR spectrum of Fe_3_O_4_@SiO_2_-Cl (Figure 4b), the absorption peaks at 2972 and 1272 cm^−1^ were ascribed to the stretching of C-H species and the symmetrical deformation vibration of Si-C species, respectively. Compared with the FTIR spectrum of Fe_3_O_4_@SiO_2_-Cl, there were no observable changes in that of Fe_3_O_4_@SiO_2_-NH_2_ in Figure 4c, but the signal of the Si-O bond vanished at 576 cm^−1^. For the FTIR spectrum of Fe_3_O_4_@SiO_2_-Br in Figure 4d, the absorption peaks at 1704, 1654 and 1339 cm^−1^ could be attributable to N-C=O species. In Figure 4d, the immobilization of BPMA on the Fe_3_O_4_@SiO_2_ surface is demonstrated by the new peak appearing at 1590 cm^−1^, which could be attributable to C=N species. Further analyses of Fe_3_O_4_@SiO_2_-BPMA were conducted by XPS, as discussed below.

Figure 5a,b show the XPS spectra of C 1s and N 1s core-levels of Fe_3_O_4_@SiO_2_-BPMA powders, respectively. As observed in Figure 5a, the C 1s core-level spectrum of Fe_3_O_4_@SiO_2_-BPMA can be curve-fitted into five peak components with binding energies of 287.9, 286.0, 285.3, 284.7 and 284.1 eV, corresponding to C=O, C-N, C-C, (C_6_H_5_-) and C-Si bonds, respectively. Moreover, the N 1s core-level spectrum in Figure 5b can be curve-fitted into three peak components with the binding energies of 401.1 eV for pyridinic-N species, 399.8 eV for N-C species and 399.0 eV for N-C=O species, respectively. Combined with the results of FTIR analyses in Figure 4, it can be concluded that BPMA was successfully bonded onto the Fe_3_O_4_@SiO_2_ surface through a series of successive coupling reactions.

The magnetic properties of Fe_3_O_4_@SiO_2_ and Fe_3_O_4_@SiO_2_-BPMA powders were investigated using a VSM at 300 K. As observed in Figure 6, the saturated magnetization of Fe_3_O_4_@SiO_2_ and Fe_3_O_4_@SiO_2_-BPMA powders were 38.1 and 20.1 emu/g with neither remanence nor coercivity, indicating their superparamagnetism. Compared with Fe_3_O_4_@SiO_2_, the decrease in saturated magnetization for Fe_3_O_4_@SiO_2_-BPMA powders was attributed to the contribution of the non-magnetic grafted organic molecule on the Fe_3_O_4_@SiO_2_ surface. The insets in Figure 6 show the photographs of the magnetic response using a magnet and re-dispersion by shaking the Fe_3_O_4_@SiO_2_-BPMA suspension. From the Figure 6 insets, it can be seen that Fe_3_O_4_@SiO_2_-BPMA particles were well dispersed in an aqueous solution, and these particles were quickly and completely separated from solution under the attraction of an external magnet within about 30 seconds in this work. Moreover, the re-dispersion of Fe_3_O_4_@SiO_2_-BPMA in aqueous solution occurred simply and quickly through slight shaking once the external magnet was removed. This demonstrates that the as-synthesized Fe_3_O_4_@SiO_2_-BPMA possesses strong magnetic sensitivity, which could make recycling easy and feasible. 

To evaluate the photoresponse of the Fe_3_O_4_@SiO_2_-BPMA nanosensor for heavy metal ions such as Cr^3+^, Cd^2+^, Hg^2+^ and Pb^2+^ ions, the fluorescence changes to various heavy metal ions with equal concentrations were measured using a 352 nm fluorescent lamp. The top of Figure 7 shows the changes in color of the Fe_3_O_4_@SiO_2_-BPMA suspension upon the addition of Me^n+^ (Me^n+^ = Cr^3+^, Cd^2+^, Hg^2+^ and Pb^2+^) illuminated by natural light; the Fe_3_O_4_@SiO_2_-BPMA suspension had no significant changes in the appearance of color after the addition of Cr^3+^, Cd^2+^, Hg^2+^ and Pb^2+^ ions in comparison with that of only Fe_3_O_4_@SiO_2_-BPMA (blank in Figure 7, top). The bottom of Figure 7 shows the eye-perceived fluorescence changes in Fe_3_O_4_@SiO_2_-BPMA toward Cr^3+^, Cd^2+^, Hg^2+^ and Pb^2+^ ions through irradiation with a 352 nm ultraviolet lamp. The considerable fluorescence enhancements can be observed by the naked eye in the vial with the Cd^2+^ ion, and the detection limit for Cd^2+^ ions reached up to 8.0 × 10^−7^ mol/L. In contrast, no significant fluorescent changes in emission were observed for the other vials containing Cr^3+^, Hg^2+^ and Pb^2+^ ions. However, the coordination diagrammatic sketch of Fe_3_O_4_@SiO_2_-BPMA with Cd^2+^ is shown in Figure 8. Upon the addition of the Cd^2+^ ion, the grafted tridentate receptor BPMA molecule coordinated with the Cd^2+^ ion, which decreased the electron-donating ability of the nitrogen atom from BPMA during the photo-induced charge transfer process, and thus, significant fluorescence enhancement occurred [26,27,28]. 

The adsorption capacity of Fe_3_O_4_@SiO_2_-BPMA nanocomposites toward the Cd^2+^ ion was evaluated at room temperature without pH pre-adjustments. Figure 9 shows the time-dependence of adsorption profiles of the Cd^2+^ ion onto Fe_3_O_4_@SiO_2_-BPMA. As observed, Fe_3_O_4_@SiO_2_-BPMA exhibited rapid adsorption to Cd^2+^ ions within 20 min, and the adsorption efficiency within 20 min reached 90.5%. Moreover, the adsorption reaction was mostly complete within 30 min, and the adsorption efficiency within 30 min reached 99.3%. As a comparison, the adsorption capacity of Fe_3_O_4_@SiO_2_ without BPMA modification was also tested, and the adsorption efficiency within 60 min was below 5.0% under the same conditions. This indicates that the absorption of the Cd^2+^ ion was mainly attributed to the grafted BPMA molecule, not Fe_3_O_4_@SiO_2_ particles.

Figure 10 shows the effects of Cd^2+^ initial concentration on the adsorption efficiency and adsorption amount of the Fe_3_O_4_@SiO_2_-BPMA adsorbent. As observed, the adsorption efficiency decreased with the increasing Cd^2+^ initial concentration, the decrease being especially sharp when the Cd^2+^ initial concentration was higher than 300 mg/L, which could be due to the saturation of the Fe_3_O_4_@SiO_2_-BPMA adsorbent. The adsorption efficiency remained at more than 94.0% when the Cd^2+^ initial concentration was lower than 300 mg/L. However, the adsorption amount increased with the increasing Cd^2+^ initial concentration, and the increasing trend receded gradually. The mass of the Fe_3_O_4_@SiO_2_-BPMA adsorbent was certain, and the relative adsorption sites gradually decreased with the increasing Cd^2+^ initial concentration and finally reached an adsorption–desorption equilibrium between the Cd^2+^ ion and the adsorbent. The saturated adsorption amount (*q*_m_*,* mg/g) of Cd^2+^ ions can be evaluated by the Langmuir isotherm model using Equations (2) and (3).
(2)$$q_{e}=\frac{(C_{0}-C_{e})V}{m}$$
(3)$$\frac{Ce}{q_{e}}=\frac{1}{q_{m}}C_{e}+\frac{1}{K_{L}q_{m}}$$
where *C*_0_ (mg/L) is the initial concentration of Cd^2+^ ions, *C*_e_ (mg/L) is the concentration of Cd^2+^ ions at equilibrium, *m* (g) is the mass of Fe_3_O_4_@SiO_2_-BPMA powders, *V* (L) is the volume of Cd^2+^ aqueous solution, *q*_e_ (mg/g) is the equilibrium adsorption amount of Fe_3_O_4_@SiO_2_-BPMA, *q*_m_ (mg/g) is the saturated adsorption amount of Fe_3_O_4_@SiO_2_-BPMA, and *K*_L_ is the Langmuir adsorption constant. The Langmuir linear fit of the adsorption of Cd^2+^ ions onto Fe_3_O_4_@SiO_2_-BPMA is shown in Figure 11. The corresponding Langmuir parameters calculated at room temperature were as follows: *q*_m_ = 342.5 mg/g and *K*_L_ = 0.5478. Moreover, a high associated correlation coefficient *R*^2^ of 0.9991 was obtained, indicating that the Langmuir isotherm model was a good fit for modeling the adsorption of Cd^2+^ ions onto Fe_3_O_4_@SiO_2_-BPMA.

## 4. Conclusions

A sensitive fluorescence sensor based on functionalized Fe_3_O_4_@SiO_2_ nanospheres was used for simultaneously detecting and removing Cd^2+^ ions. The as-synthesized Fe_3_O_4_@SiO_2_-BPMA underwent an eye-perceived fluorescence enhancement induced by Cd^2+^ ions. Meanwhile, Fe_3_O_4_@SiO_2_-BPMA exhibited rapid and effective adsorption toward the Cd^2+^ ion, and the adsorption reaction was mostly complete within 30 min. The Cd^2+^ adsorption capacity of Fe_3_O_4_@SiO_2_-BPMA was determined by fitting the experimental data with the Langmuir model, and the saturated adsorption amount was 342.5 mg/g at room temperature. Moreover, Fe_3_O_4_@SiO_2_-BPMA showed superparamagnetic properties with a saturated magnetization of 20.1 emu/g, which could help to separate these particles after capturing Cd^2+^ ions. The nanocomposites presented here have great potential applications for the naked-eye identification, adsorption and separation of the Cd^2+^ ion.

## Figures and Tables

**Figure 1 materials-13-03678-f001:**
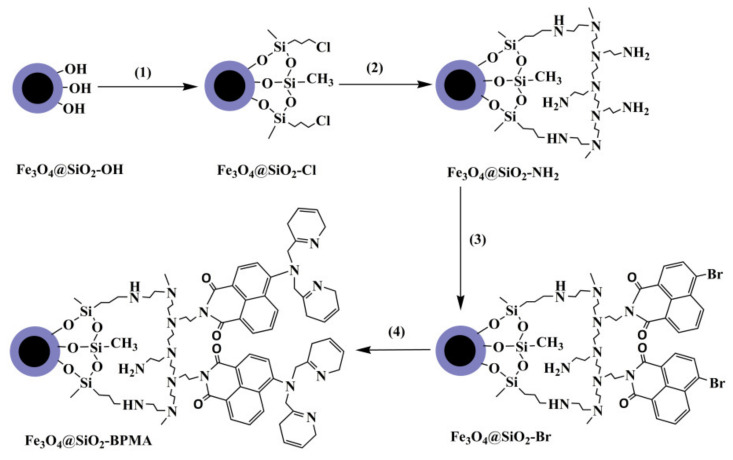
Synthesis of bis(2-pyridylmethyl)amine (BPMA)-functionalized Fe_3_O_4_@SiO_2_ by a “grafting-from” approach.

**Figure 2 materials-13-03678-f002:**
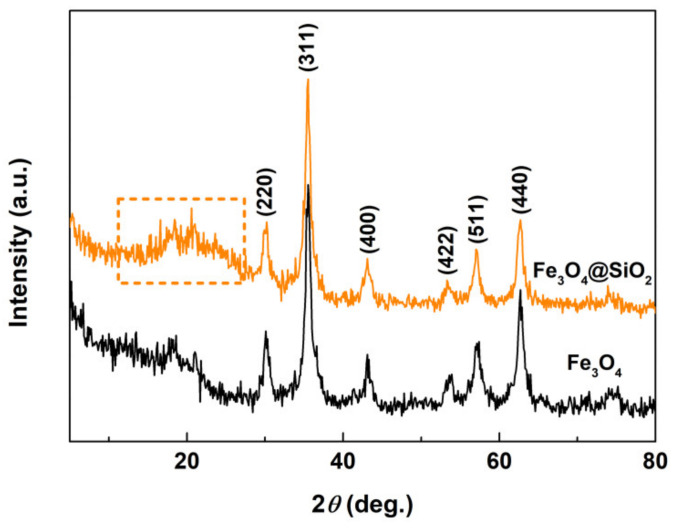
XRD patterns of Fe_3_O_4_ and Fe_3_O_4_@SiO_2_ powders.

**Figure 3 materials-13-03678-f003:**
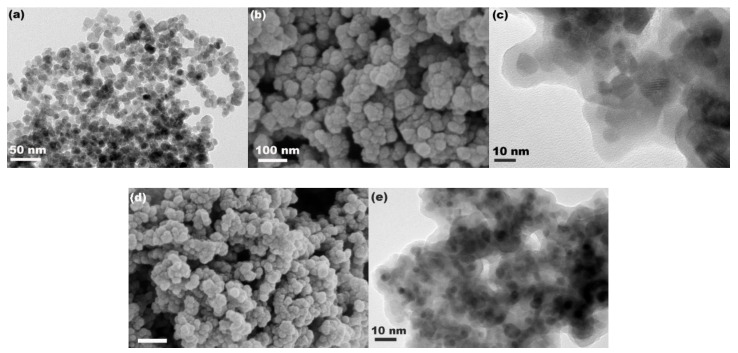
(**a**) TEM image of Fe_3_O_4_ particles, (**b**) SEM and (**c**) TEM images of Fe_3_O_4_@SiO_2_ particles, (**d**) SEM and (**e**) TEM images of Fe_3_O_4_@SiO_2_-BPMA particles.

**Figure 4 materials-13-03678-f004:**
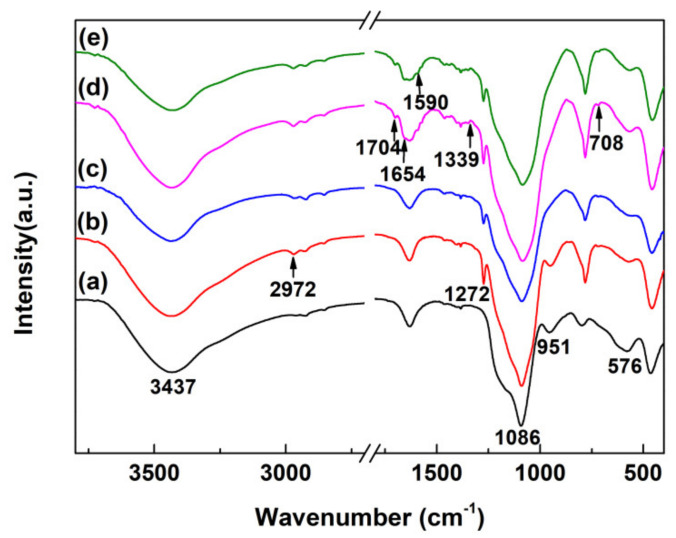
FTIR spectra of (**a**) Fe_3_O_4_@SiO_2_, (**b**) Fe_3_O_4_@SiO_2_-Cl, (**c**) Fe_3_O_4_@SiO_2_-NH_2_, (**d**) Fe_3_O_4_@SiO_2_-Br and (**e**) Fe_3_O_4_@SiO_2_-BPMA powders.

**Figure 5 materials-13-03678-f005:**
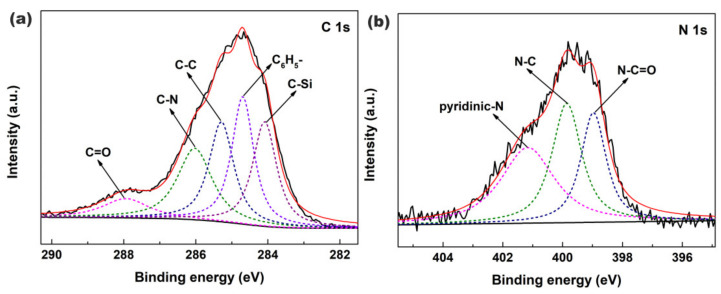
XPS spectra of (**a**) C 1s and (**b**) N 1s core-levels of Fe_3_O_4_@SiO_2_-BPMA powders.

**Figure 6 materials-13-03678-f006:**
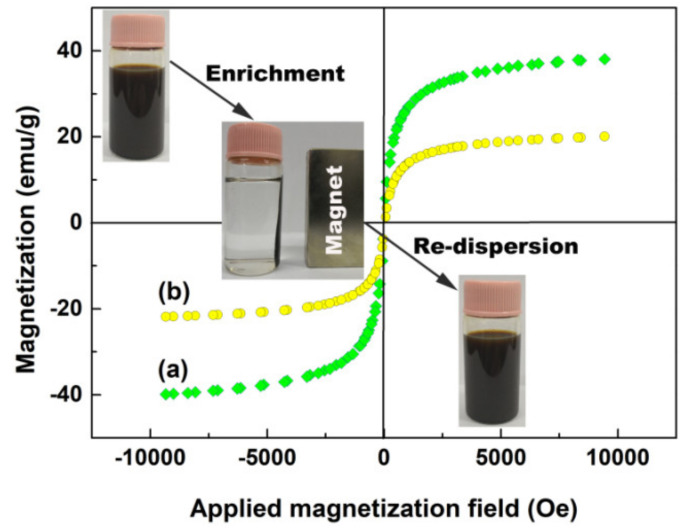
Magnetization hysteresis loops of Fe_3_O_4_@SiO_2_ (**a**) and Fe_3_O_4_@SiO_2_-BPMA (**b**) powders (insets are the Fe_3_O_4_@SiO_2_-BPMA suspension, [Fe_3_O_4_@SiO_2_-BPMA] = 1.25 g/L, *V* = 8.0 mL, aqueous medium).

**Figure 7 materials-13-03678-f007:**
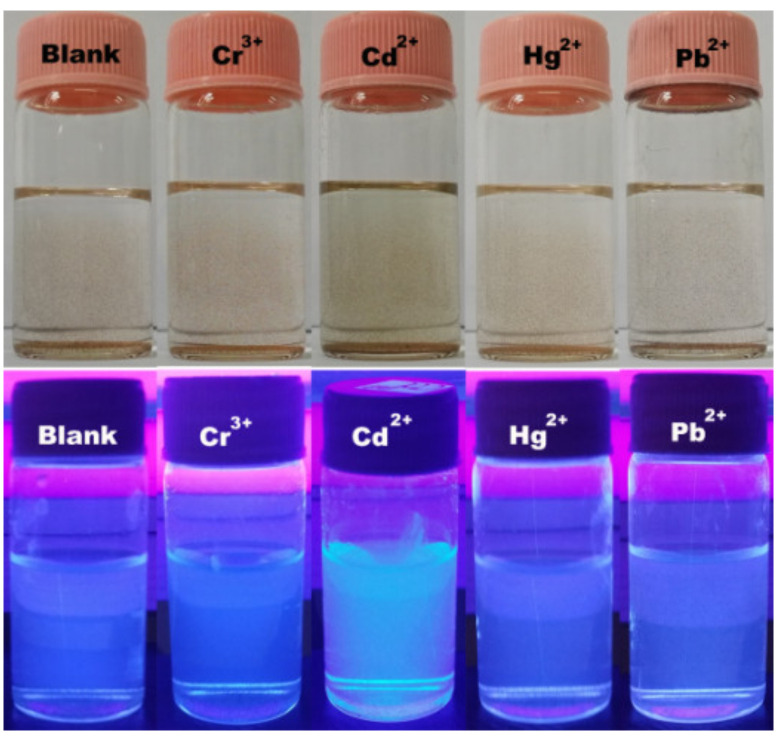
Changes in color (**top**) illuminated by natural light and fluorescence (**bottom**) illuminated with a fluorescent lamp (λ_ex_ = 352 nm, 8 W) of Fe_3_O_4_@SiO_2_-BPMA upon the addition of Me^n+^ (Me^n+^ = Cr^3+^, Cd^2+^, Hg^2+^ and Pb^2+^) ([Fe_3_O_4_@SiO_2_-BPMA] = 0.1 g/L; [Me^n+^] = 10^−5^ mol/L; *V* = 6 mL, *V*_acetonitrile_/*V*_water_ = 4/1).

**Figure 8 materials-13-03678-f008:**
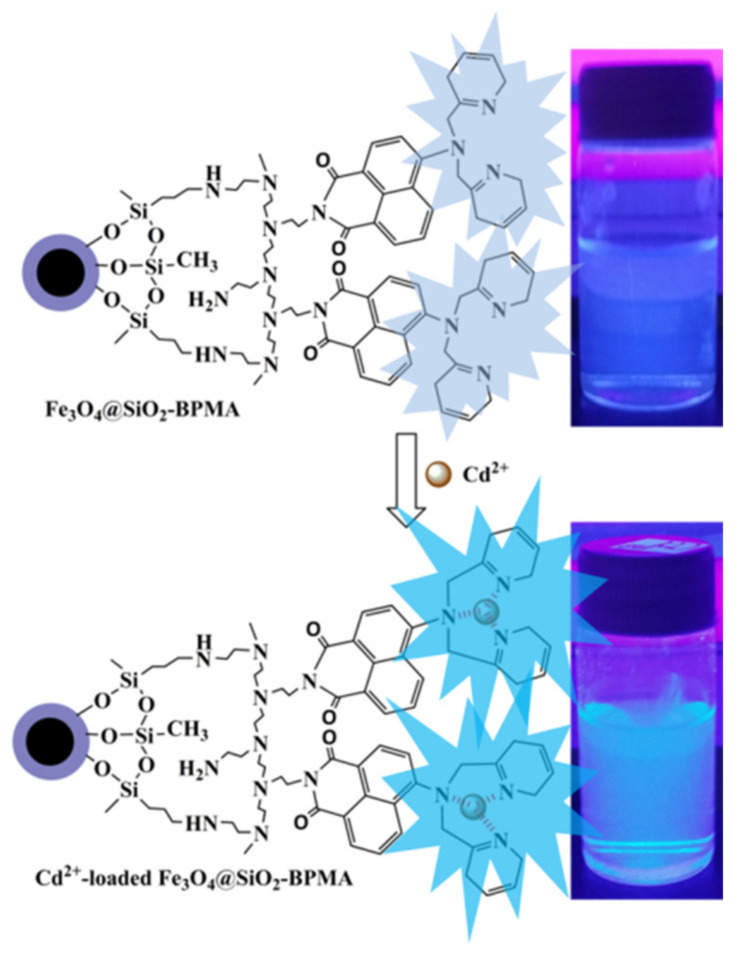
Binding mechanism of Fe_3_O_4_@SiO_2_-BPMA toward Cd^2+^ ions.

**Figure 9 materials-13-03678-f009:**
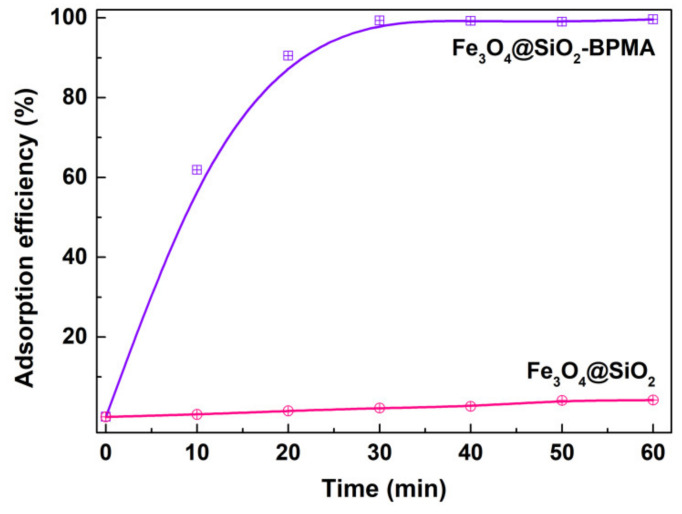
Time-dependence of adsorption profiles of Cd^2+^ ion onto Fe_3_O_4_@SiO_2_ and Fe_3_O_4_@SiO_2_-BPMA powders (adsorption conditions: room temperature; no pH pre-adjustments; [Fe_3_O_4_@SiO_2_-BPMA] = 1.0 g/L; [Cd^2+^] = 200 mg/L; *V* = 50 mL; *t* = 0–60 min).

**Figure 10 materials-13-03678-f010:**
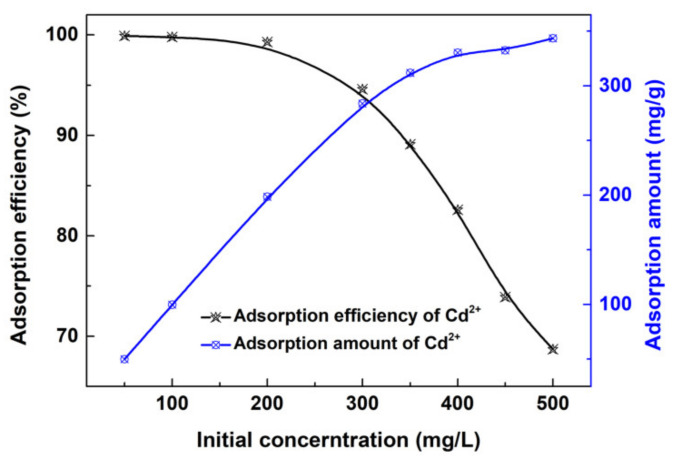
Effects of Cd^2−^ initial concentration on the adsorption efficiency and adsorption amount (adsorption conditions: room temperature; no pH pre-adjustments; [Fe_3_O_4_@SiO_2_-BPMA] = 1.0 g/L; [Cd^2+^] = 50–500 mg/L; *V* = 50 mL; *t* = 30 min).

**Figure 11 materials-13-03678-f011:**
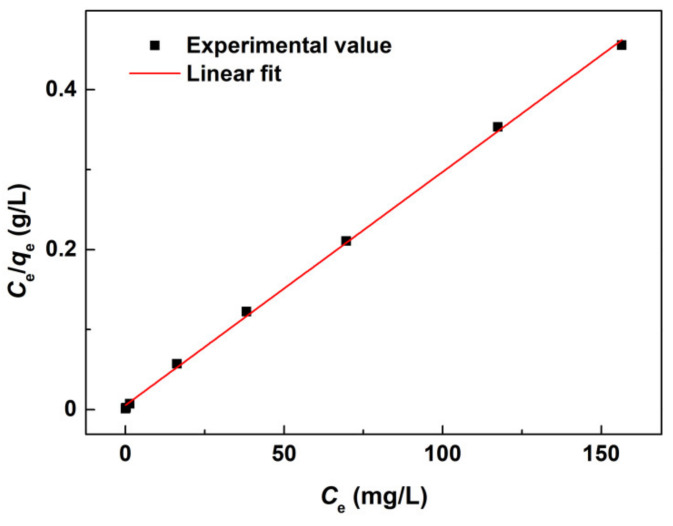
Langmuir linear fit of Cd^2+^ adsorbed onto Fe_3_O_4_@SiO_2_-BPMA.

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
