# Peer review of "A Nanosensor for Naked-Eye Identification and Adsorption of Cadmium Ion Based on Core–Shell Magnetic Nanospheres"

_materials, 2020, doi:10.3390/ma13173678_

Round 1
Reviewer 1 Report
The toxicity of Cd both for the environment and the human being makes the topic of this paper of great interest in principle. The quality of the English language and in particular of punctuation use rests merit and readibility. This shoould be carefully checked in my opinion. Regarding the technical contents of the manuscript, I would like to point out the following:
- Line 43: Hg2+ detection? Misprint I suppose
- Why precisely BPMA for functionalization? This should be more carefully justified
- Fig. 3b and 3c: It seems that the magnetite/silica composites are highly aggregated. Some comments on this wold be adequate. Also, is there any effect on the particle size or stability upon functionalization with BPMA?
- L. 151 ff: why is the system specifically sensitive to Cd2+ and not to the other cations?
- Fig. 6: is it possible to quantify fluorescence, beyond naked-eye observations?
- Fig. 8 How is Cd2+ adsorption evaluated?
- Lines 204 ff: This paragraph regarding magnetic properties should be moved to the characterization part, as it is completely out of place here, at the end of the paper and not related to immediately previous contents
Reviewer 2 Report
Xu and co-workers submitted the paper entitled “A Magnetic Nanosensor for Naked Eye Identification and Adsorption of Cd2+ Ion Based on Core-Shell Fe3O4@SiO2 Nanospheres” for possible publication in “Materials”. In this paper, they synthesized Fe3O4@SiO2 core-shell nanoparticles and functionalized them with Bis(2-pyridylmethyl)amine (BPMA). The as-synthesized Fe3O4@SiO2-BPMA nanoparticles were used to detect Cd2+ ion by selective fluorescence enhancement. The nanoparticles showed a high absorption coefficiency of 99.3% with a saturated adsorption amount of 342.5 mg/g based on Langmuir linear fitting. Moreover, Fe3O4@SiO2-BPMA displayed superparamagnetic properties with saturated magnetization of 20.1 emu/g. Overall the manuscript was properly organized and written, however, there are still some missing information and queries that needs to be provided and answered before its acceptance.
- Binding site and mechanism need further experimental proof, or at least DFT calculations to confirm.
- Limits of detection, selectivity, or interference data were missing.
- Real applications, including testing in tap water, sea water, should be provided?
- In Figure 10, curves (a) and (b) are not clearly labelled.
Reviewer 3 Report
Manuscript title: A Magnetic Nanosensor for Naked Eye Identification and Adsorption of Cd2+ Ion Based on Core-Shell Fe3O4@SiO2 Nanospheres
This manuscript described the preparation and characterisation, and the application of Fe3o4@SiO2 based nanoparticle for the Cd adsorption and detection. The NPs were prepared by stepwise procedure starting form Fe3o4, coating with silica and then the modification of NPs with naphthanilime based dye for Cd capturing and sensing. After binding of Fe3o4@SiO2-BPMA with Cd, the fluorescence signal was switched on, which could be used for the Cd detection. This is an interesting work, could be recommended for publishing after addressing following questions:
- The manuscript is suggested to be carefully checked to avoid typos. Please include full name of Cd in the title; Fe3o4@SiO2 could be removed from title.
- Figure showed the procedure for the preparation of Fe3o4@SiO2-BPMA, please include characterisation data, such as FTIR.
- The size distribution of Fe3o4@SiO2-BPMA is suggested to be characterised by DLS. And TEM, SEM characterisations for the NPs are also suggested.
- The specific fluorescence “OFF-ON” response is suggested to be examined by including Zn2+ and Cu2+ in Figure 6.
- The changes of absorption spectra and emission spectra are suggested to be added.
Round 2
Reviewer 2 Report
The revised manuscript can now be accepted.
Reviewer 3 Report
The manuscript is now recommended for publishing. All questions have been well addressed.